# Pathways to Livable Relocation Settlements Following Disaster

**Shaye Palagi \*** and **Amy Javernick-Will**

Department of Civil, Environmental and Architectural Engineering, University of Colorado Boulder, Boulder, CO 80309, USA; amy.javernick@colorado.edu
**\*** Correspondence: shaye.palagi@colorado.edu; Tel.: +1-406-788-1958

**Abstract:** Mass relocation—the transfer of communities to new housing developments—is often implemented following disasters, despite criticism that past projects have not created livable communities for residents. Livable relocation communities are those where residents experience quality housing, utilities, social infrastructure, neighborliness, safety, and a sense of permanence. Numerous conditions may support livability, such as site location, community involvement, and processes of managing construction and beneficiary transfer. We evaluated relocation communities in Tacloban City, Philippines, applying Qualitative Comparative Analysis to identify pathways, or combinations of conditions, that led to built and societal livability. We found pathways to livability generally differed between government and non-government developed sites, with the former benefiting from a slower pace and standard permitting procedures, and the latter by building fast and using scale and need to prompt improved services. An unexpected combination emerged as a pathway to societal livability—being remote and comprised of households originally from a mix of different communities—revealing a new narrative for positive social outcomes in relocation. Three conditions emerged as necessary for achieving overall livability: fast construction, full occupancy, and close proximity to an economic and administrative center. This analysis demonstrates necessary conditions and pathways that implementing agencies can reference in their quest to create livable relocation communities.

**Keywords:** post-disaster; relocation; livability; Typhoon Haiyan

---

## 1. Introduction

Post-disaster mass relocation, the transfer of entire communities to new housing projects, is often selected as a risk reduction and recovery strategy following large-scale disasters. Relocation has been implemented globally following many types of hazard events, such as post-earthquake relocation in 1970 in Turkey and post-tsunami relocation in 2004 throughout Indian Ocean countries [1,2]. Mass relocation generally occurs in low- and middle-income countries where large swaths of the disaster-affected population often do not have ownership or formal tenure to their homes or the land. Most often, mass relocation has been ordered by governments and managed as a collection of large-scale housing developments, institutionally-funded and contractor-built, with limited participation from intended occupants [3]. Households may be given their relocation houses for no to little cost, but often with limited official ownership, either over physical assets or the decisions that preceded them. In the context of this paper, relocation is confined to such government-mandated, organizationally-driven housing development projects intended for low-income households, most with no or informal prior tenure security, displaced by a natural hazard.

Despite their persistent selection as a risk reduction strategy, relocation projects themselves are highly criticized for lacking basic facilities and utilities, severing households from vital social

or economic ties, and even exposing households to additional hazards [4,5]. Occasionally, internal inadequacies, or an inability to access income, opportunities, and daily necessities, become so impactful that households decide to abandon their provided houses [6,7]. We posit that abandonment of relocation projects is an issue of livability, i.e., the relocation community's quality of life. In this research, we view livable relocation communities as those residents can and want to preside in long-term. In livable relocation sites, residents enjoy perceived quality housing, reliable utilities, nearby social infrastructure, neighborliness, safety, and a sense of permanence. Trends suggest relocation will persist after disasters, for example the governments of both Mozambique and Zimbabwe are planning significant relocation developments following a series of cyclones in 2019 [8]. Thus, there is a need to better qualify both livability and the relevant factors facilitating it. In this study, we aimed to outline what is unique about 'livability' in post-disaster relocation sites, enumerate the conditions that may contribute to livable communities, and uncover causal relations between the two. By discerning not just failures at relocation sites but how livability might be facilitated, implementors of future relocation projects can better devise programs capable of stimulating livable relocation communities.

## 2. Background

Most metrics of livability encompass human needs, ranging from securing basic infrastructure to the sense of belonging or opportunity [9]. Further, context matters, and aspects of habitability that are taken-for-granted in some contexts may warrant specific attention in others [10]. In high income countries livability measures tend to focus on housing, public transit, and open and recreational space, while considerations in low- and middle-income countries may necessitate focus on potable water, labor and tenure rights, waste management, and hazard susceptibility [11]. Our study focuses on livability to assess and improve relocation projects. Currently, however, livability metrics for these projects do not exist and must be modified from metrics applied to other contexts and scales. For our purposes, we draw from literature of urban livability based on current resident quality of life at the local scale (i.e., neighborhood, rather than city, regional, or country scale) and adapt insights to relocation communities in a lower middle-income country, the Philippines.

Broadly defined, urban livability is "the sum total of the qualities of the urban environment that tend to induce in a citizen a state of well-being and satisfaction" [12] (p. 13). Some scholars describe livability as a discernable objective quality of life in a given locale [13]. Population density, commute time, and air quality are examples of objective dimensions of livability [14]. However, especially at the local-level, others contend that livability is relative and best understood from the vantage point of community residents [15,16]. For instance, del Rio et al. (2012) began their study of livability in a Brazil favela by first asking what residents themselves considered essential to a livable community [17]. They found residents were generally satisfied with their housing, but that community cohesion forged in public squares and streets were the drivers of livability [17]. While there is perhaps more variety than uniformity among definitions of livability, nearly all invoke the experience of residents themselves. A key theme across the breadth of definitions is "that it is inherently anthropocentric – livability is a reflection of 'quality of life', 'wellbeing' and/or the satisfaction of the needs of 'the people'" [18] (p. 123). We build from the experiential, community-grounded tradition of urban livability studies and accordingly use resident judgements to assess community-level relocation livability. Perceived livability is especially critical in fledgling relocation projects with the potential to either become increasingly livable, or not.

We measured perceived livability as an outcome in post-disaster relocation communities. Livability is not a guaranteed outcome. Instead, we hypothesized project conditions, including the pace, scale, location, level of community involvement, and other dynamics support or hinder community livability. Below, we synthesize literature to both define relocation livability and postulate project conditions that contribute to livability.

*2.1. Identifying and Measuring Livable Relocation Communities*

There is no unified definitions of 'successful or failed' post-disaster relocation communities [7]. However, relocation case studies and disaster literature evaluating local-level recovery both signal intended aspects of livable relocation projects. For example, Jordan and Javernick-Will's (2013) comprehensive literature review of community recovery indicators identified quality housing, functional services and facilities, and reliable transportation [19]. Others used similar concepts when comparing resettled communities, including house quality, social infrastructure, social cohesion, site planning, and livelihood opportunities [20].

The diversity of recovery metrics is matched by similarly diverse metrics of livability. For instance, Wei and Chiu (2018) divided their study into the built environment and its available resources as well as the sense of community derived from living in a particular place [21]. Slightly more nuanced, Leby and Hashim (2010) summarized dimensions of livability into physical, functional, social and safety [22]. While researchers vary in exactly how they decompose livability and organize its sub-parts, there are common themes across studies. In particular, most past research has classified aspects of livability into tangible and intangible measures [15,18]. We follow this divide by examining aspects of livability into two dimensions: built and societal livability. Building onto our earlier definition that livable relocation communities are those communities where residents can and want to preside within long-term, we argue that they can stay because their built environment is functional and meets their needs, and they want to stay because they are members of a gratifying society.

Built livability includes the communal, housing, and civil infrastructure that contribute to a community's quality of life. Research has found built amenities, including electricity, roads, and schools, to be fundamental to enduring relocation projects [2]. In this research, we select three sub-constructs of the built environment to assess built livability: civil infrastructure, housing, and proximity of communal infrastructure and necessities. Civil infrastructure services include potable and domestic water, safe sewage disposal, electricity, and passable road networks, all with enough capacity and reliability to sustain the community. Housing measures include perceived satisfaction and quality [23]. With growing recognition that relocation requires more than housing, communal infrastructure (e.g., schools and hospitals) have accompanied recent projects [24]. Additionally, quality of high-density living can be improved by access to local supply, such as markets and stores where residents can purchase everyday needs and avoid burdensome trips for small necessities [25]. In this study, built livability is assessed as a composite of infrastructure, housing, and accessible services.

In comparison, societal livability addresses the social and psychological sense of community, safety, and permanence [26]. Social connections are a form of a capital, and relocated residents can benefit from strong relations with each other (bonding social capital) and with local administrators (linking social capital) [27]. To be livable, relocation communities must also be safe. Women and adolescents in high-poverty communities—relocation communities are inherently high-poverty as they are commonly intended for the urban poor—may be at an increased risk of violence [28,29]. However, connections among neighbors can buffer negative consequences [30]. Finally, the transition from informal settlements should come with not only housing, but security of tenure [14]. Past research has demonstrated relocation housing projects have been developed without any initial plan to provide tenants ownership, or to provide conditional tenure based on tenant behavior [31]. Secure tenure is crucial for a sense of secure community membership, those without legal permanence are not fully livable. Here, societal livability is assessed as a composite of bonding social capital, linking social capital, safety, and tenure.

Deconstructing relocation livability into built and societal dimensions allows us to separately examine not only outcomes but also the causal pathways to each outcome. Separating livability into built and societal dimensions also leaves room for the possibility that a community may achieve partial livability (one or the other, not both) or that built and societal livability may be induced by differing causal processes.

## 2.2. Theorized Contributors to Relocation Livability

What enables or hinders livability in relocation communities is contested. Some debates are not relocation-specific, but central in all post-disaster reconstruction efforts, such as the conflict between rapid rebuild and deliberate planning [32]. Even so, relocation projects are distinctive; unlike in-situ reconstruction, there is no preexisting image of the community to catalyze rebuilding. In practice, agency-driven, as opposed to owner-driven, approaches remain the norm [33,34]; thus, we focused on the following list of theorized contributors to livability that may vary between agency-driven projects.

### 2.2.1. Site Selection and Proximity

Site selection is critical for project outcomes and often blamed for failed relocation, in part because sites are rarely selected based on the needs of intended inhabitants [35,36]. As Oliver-Smith (1991) noted, "sites for resettlement after disaster are often chosen with factors other than the welfare and development of the population in mind" [2] (p. 16). Relocation sites are occasionally selected because they are unexposed to hazards, at least the hazard type that prompted relocation, and expected to impose minimal negative repercussions on residents' quality of life. In practice, however, sites are most commonly selected for favorable topography, rapid or expansive development, easy and inexpensive acquisition, ignorance, or a lack of concern for ecological or economical concerns [2].

The dominant critique of relocation is distance. Past research resoundingly criticizes relocation projects for being too far from economic centers or original communities and attributes the distance as a cause of low resident satisfaction, poor civil infrastructure services, and increased poverty [37,38]. Even when relocated to the urban periphery, transportation costs increase and can significantly impact a family's budget [39]. Conversely, short relocation distances are credited as a contributor to recovery and resilience [40,41]. In addition, proximity to urban centers has been positively correlated with high social capital [42]. Recognizing the plentiful critiques against distant relocation, we hypothesize that locating projects near economic hubs and old communities is necessary for livable relocation communities.

### 2.2.2. Construction

Prior research suggests the pace, (how fast the project is constructed), scale (how many houses are to be constructed), and quality (standard of workmanship) of construction will affect outcomes at relocation sites. Speed versus deliberation is one of the core trade-offs in post-disaster recovery and relocation [43]. With speed, citizens can enjoy minimal disruption and a rapid return to normalcy. In relocation projects, fast construction means that relocated households are able to quickly move in and form social ties with their new community. Additionally, pace is often viewed as a proxy for how well the government prioritizes recovery, and overall satisfaction with government recovery efforts can decrease if relocated communities have to wait a long time to move in [44]. However, rapid development has been targeted as a cause of both poor housing and civil infrastructure [45,46].

The appropriate size for relocation projects is often implicit in discussions of pace; smaller sites can be developed easier and quicker. However, rushed development can have negative consequences, particularly in construction of the built environment. Contractors balance pace with quality, and workmanship may lax with hurried development [47]. In construction booms, designs are often rushed, contracts are entered with incomplete information, and errors are made [48]. In addition, government agencies can struggle to meet the permitting demand, which may lead others to blame slow progress on bureaucratic red-tape. Like pace, there are also potential drawbacks to numerous small-sized developments, which may lead to inefficiencies in managing the overall recovery need or establishing an integrated new urban area.

Researchers have observed that relocation development generally takes longer than in-situ reconstruction and unfolds over several years [49,50]. For instance, after the 2010 Chilean earthquake, government-funded housing progressed over the course of four years, where only 5%, 10%, and 68% of new construction was completed within the first 1, 2, and 3 years after the earthquake, respectively [51].

As a result, households have waited for years in transitional housing [50] before being relocated. Given the drawbacks and benefits to both fast and slow construction, the pace and scale of relocation construction should be considered and analyzed alongside other project elements.

### 2.2.3. Community Involvement

The importance of participation is prominent in literature addressing both in situ reconstruction and relocation, but relocation projects are less likely to include participation [34,52]. For instance, researchers studying Chilean post-tsunami relocation outcomes attributed differences, in part, to variations in community involvement [53]. When decision making excludes local community processes, communities can feel isolated from the decisions impacting them [44,54]. Past scholars have indicated the most important decision in which to involve to-be-relocated households is selecting where the new relocation site should be developed [6]. Other critical decisions include house design, material selection, community layout, and procedures for implementing the transfer [55,56]. Despite the value of up-front participation in early decisions, participatory approaches are often limited to the inclusion of households in construction [23,57]. Participation in construction generally occurs through sweat equity, which may be required for a new house in lieu of financial equity. Yet capping participation to sweat equity both undervalues the contributions households can make and forces them to abandon remuneration activities to meet labor requirements [52]. While it is apparent scholars would advise participation to support livability, they would also caution that the degree of participation, and participation in what element (e.g., early decision making), matters.

### 2.2.4. The Dynamics of Transfer

The transfer of households into relocation sites can occur in many ways. For instance, all residents can originate from the same community, or not. They may first reside in transitional shelter, or not. They may be moved all at the same time, or not. As a result, transfer dynamics require further study. The majority of researchers have stressed the importance of keeping communities intact throughout the relocation process in order to preserve existing social ties [58,59]. Jumbling relocated households both severs them from their prior social relations and burdens them with forging new ones, and potentially weakens their social capital [60,61]. This may be especially if their social networks are upended numerous times as they transfer from their original communities, to transitional housing, and then into permanent housing. Since constructing entirely new housing developments takes time, beneficiaries may be provided less-durable, faster-constructed transitional shelters prior to relocation [62–64].

Additionally, cohesive transfer, or the simultaneous movement of households into their relocation community, is an important consideration. Cohesive transfer is a taken-for-granted expectation among disaster scholars, whereby households transferred from a singular original community, are assumed to be transferred simultaneously. As our context will soon bear out, transfers can be more disjointed in space and time than generally recognized in literature. Programmatically, organizations managing relocation may be limited in how many households they can facilitate transferring in a given timeframe [65]. Relatedly, not all relocation sites are immediately occupied to their intended capacity. When this occurs, low occupancy may reduce infrastructure service delivery and increase deterioration as infrastructure goes unused. Similarly, decision makers, who need to prioritize resources, may divert resources to where they will affect the most families, which is at sites with high occupancy. A parallel can be made with shrinking cities. Depopulation has been linked with building and infrastructure decay, social upheaval, and ineffectual bureaucracies [66–68]. Although a different context, the affiliated consequences of under-population may be similar. Thus, all considered, we accounted for the homogeneity of originating communities, whether households transferred from transitional sites, the cohesiveness of transfers through time, and whether sites reach full occupancy as various dynamics of relocation transfer.

In summary, no condition theorized to contribute to livability is entirely independent of another. For instance, construction pace is not independent of scale—a greater housing need and larger projects

generally require more time to complete. Similarly, highly participatory project planning is easier to facilitate in one neighborhood that will transition homogeneously into a new relocation site that across geographically diverse households. In order to consider these theorized combinations of conditions, we employed fuzzy-set Qualitative Comparative Analysis (fsQCA), a method designed to investigate how relationships among multiple conditions ultimately lead to an outcome.

## 3. Methods

We assessed livability in thirteen post-disaster relocation communities in the Philippines. To determine what combination of conditions enabled the outcome of livable relocation sites we used the configurational technique fsQCA. We discuss our context, data collection and fsQCA below, including the justification for methodological decisions. Supplementary Material is provided with this article containing comprehensive details for condition calibration (SI-1), outcome calibration (SI-2), and analysis (SI-3).

### 3.1. Context

We investigated livable relocation communities in the context of Tacloban City, Philippines, which was devastated by Typhoon Haiyan in November 2013. Although the typhoon afflicted multi-region damage, Tacloban City was the epicenter of destruction. With a daytime population of nearly a quarter million people, Tacloban City is a bustling regional hub that faces urbanization challenges like many growing Asian cities, including seaside informal settlements [69,70]. Following Typhoon Haiyan, the city government decided to relocate informal settlers (up to 40 percent of the city's population) to the largely undeveloped northern edges of the city, and envisioned a development with a vibrant environment, social services, and livelihoods [71]. Both the central government's National Housing Authority (NHA) and numerous non-governmental organizations committed to constructing socialized housing projects, yet the NHA took on the largest share by committing to more than 14,000 houses. Iuchi and Maly (2016) traced the trajectories of households after the disaster and found that many households journeyed through multiple and diverse sheltering solutions after Typhoon Haiyan before settling into permanent relocation homes [72]. For the first two years of implementation, development was project-oriented and unintegrated. Relocation projects ranged from people-centered to completely void of beneficiary participation [34,73]. Out of the 29 relocation projects tracked by the Tacloban City Housing and Community Development Office (as of October 2017), thirteen relocation projects were selected, based on diversity in implementing agency (NGO, bilateral, or NHA management) and project conditions, which were anticipated to influence livability outcomes. Coordinated implementation for multi-site integrated services improved in 2015 and 2016 with the city-led push to develop (and secure national funding for) the Tacloban North Integrated Development Plan [74]. Although the initial city plan entailed completing the permanent relocation housing by the storm's three year anniversary, the idealized goal for Tacloban North was largely not met [71].

### 3.2. Data Collection

Data were collected in three successive fieldwork trips spanning 2016 to 2018 and totaling nine months. We collected data from dual perspectives, including decision makers and community members. We sought to understand the constraints and options facing decision makers to inform condition selection and adaptation to our context. We conducted 51 interviews with decision makers across national, regional, and local bodies as well as representatives from non-governmental and community advocacy organizations. We also attended and observed 21 meetings for coordination, planning, or reporting. In order to determine outcome metrics relevant to relocation communities, we interviewed 106 community members in eight different relocation sites and longitudinally observed construction across case sites.

Following initial qualitative fieldwork, we designed a relocation-specific survey questionnaire to use in calibrating the outcome of livability as well as several conditions posited to influence livability

(discussed below). We conducted a survey because the scale of many of the sites, some nearing 1000 households, made finding theoretical saturation with interviews alone difficult. Sample size per relocation community was determined based on Dillman's (2000) recommended calculation and based on field-validated occupancy [75]. In partnership with local assistants, we piloted and then administered the survey from October 2017 to April 2018. Surveys included 80 questions and took roughly 30 min to complete. Communities are summarized in Table 1. Pseudonyms have been used for each community. Households had been living in the relocation sites for, on average, two years by the start of the survey. At the most recently occupied site, households first moved-in ten months before the start of the survey. In all, we administered 976 surveys across the thirteen relocation case sites.

**Table 1.** Case community occupancy and survey sample size.

| Site | Project Type | Planned Occupancy | Validated Occupancy (Oct. 2017) | Sample Size | Percent Surveyed (of val. occ.) | Surveyed Women (Percent) |
|---|---|---|---|---|---|---|
| Gamay | NGO | 403 | 394 | 110 | 28 | 74.3% |
| Gandara | NHA | 1000 | 737 | 82 | 11 | 74.4% |
| Hitunlob | NGO | 503 | 450 | 81 | 18 | 76.5% |
| Lawigan | NGO | 52 | 52 | 39 | 75 | 71.8% |
| Nagaja | NHA | 1000 | 914 | 107 | 12 | 60.4% |
| Natubgan | NHA | 1000 | 923 | 80 | 9 | 60.0% |
| Oras | NGO | 100 | 76 | 44 | 58 | 75.0% |
| Pangasugan | NHA | 1000 | 830 | 76 | 9 | 68.4% |
| Sambawan | NGO | 600 | 495 | 86 | 17 | 79.1% |
| Suribao | NHA | 1000 | 300 | 73 | 24 | 54.8% |
| Ulot | NGO | 55 | 55 | 40 | 73 | 87.2% |
| Villareal | NHA | 409 | 378 | 78 | 21 | 63.6% |
| Vulcan | NHA | 584 | 488 | 80 | 16 | 65.0% |
| Total | - | 7706 | 6092 | 976 | 16 | 69.1% |

When administering surveys, assistants always emphasized that they represented independent researchers from a foreign university, without any affiliation with an NGO or government office; that participation was voluntary, confidential, and without direct benefits (i.e., we did not provide gifts); and that no questions were mandatory (respondents occasionally felt they did not have an opinion or did not feel comfortable answering). Assistants administered the survey in Waray-Waray, the local language and record responses digitally. The research was conducted following a review by the University of Colorado Boulder Institutional Review Board, Protocol 16-0245. Survey descriptive statistics have been shared with relocation site leaders and pertinent city offices to ensure findings were accessible to studied communities.

### 3.3. Determining Causality: Fuzzy-Set Qualitative Comparative Analysis (fsQCA)

To determine combinations of conditions that resulted in livable communities, we analyzed and compared our data across cases using fsQCA. Qualitative comparative analysis was born out of the assertion that case-oriented research is often naturally and verbally explained in terms of set theory, but was lacking a complementary analytical tool to explore set relationships [76]. Set theory is a mathematical means of describing collections, such as the set of all relocation projects, and their relations to each other, for instance the set of all relocation projects is a subset of all housing recovery modalities.

QCA allows researchers the analytical strength of quantitative methods while maintaining the rich case knowledge of qualitative studies. QCA relies on configuration analysis to discern causal "recipes", combinations of conditions that lead to the outcome of interest [77]. Fuzzy-set QCA (fsQCA), in particular, allows for the consideration of partial membership in higher-order, complex phenomena using fuzzy-set theory, the idea that cases can have varying degrees of membership within a set [78].

Numerically, a case that is in a given set is assigned a value of 1. Likewise, cases not in the set are assigned a value of 0. Cases with varying degrees of membership, 'partial membership', are assigned values between 0 and 1. Since we anticipated that our cases would vary in degree of membership for the outcome and causal conditions, fsQCA was analytically appropriate.

Fuzzy-set QCA is an iterative process, wherein the outcome of interest (in our case, livability) motivates the selection of cases, conditions leading to the outcome originate from both theory and case knowledge, and calibrations are open to refinement [79]. Figure 1 (adapted from Jordan et al. 2011), displays an overview of the fsQCA process, as well as the constructs forming our conditions and outcomes. The decisions made at each step and throughout iterations are essential to the presentation of the results [80]. Thus, before describing the analysis of fsQCA—the process of truth table minimization wherein relations among combined conditions and the outcome are analyzed—we first summarize how the conditions and outcomes were defined as fuzzy-sets.

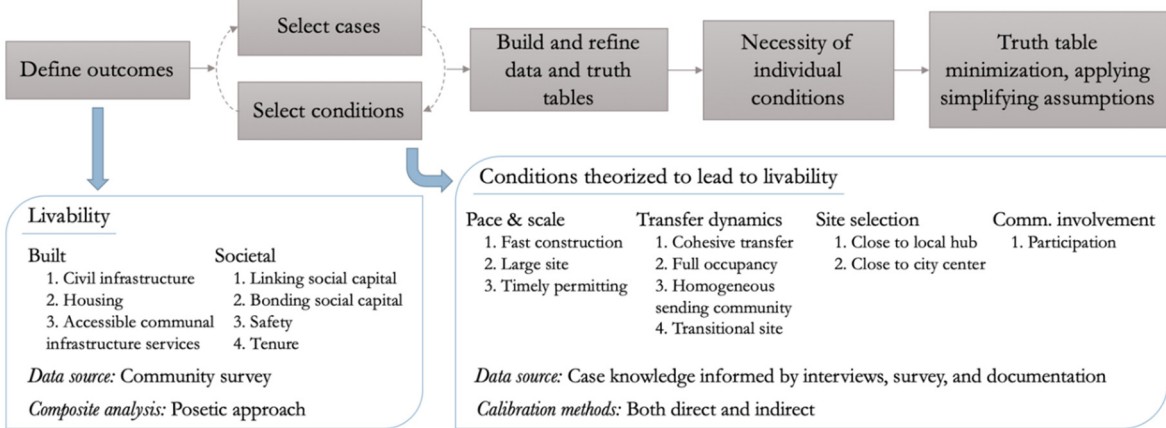

**Figure 1.** Overall fsQCA process and overview of conditions and outcomes.

### 3.3.1. Calibrating Conditions

Conditions posited to influence livability from theory were eliminated or refined through fieldwork. For example, theory suggests 'construction quality' may be an important condition for eventual livability. However, because some site owners restricted visibility during construction, workmanship quality could not be assessed systematically and contemporaneously across all sites. Because the potential proxy, final housing quality, is a component of the outcome of livability, 'construction quality' could not be included as a condition. The perspective of decision makers was critical in translating theorized conditions for livability into conditions-in-practice for Tacloban City relocation development. We targeted theorized conditions that varied among sites and adapted membership definitions to be contextually appropriate. Where applicable, we used demographic data from the survey to develop direct calibrations. Fuzzy-set membership based on direct calibrations were grounded in the specification of qualitative anchors: fully in the set (assigned a value of 1), neither in nor out of the set (assigned a value of 0.5), and fully out of the set (assigned a value of 0) [76]. Other conditions relied on indirect calibration, the determination of qualitative sets based on data with theoretical or case-based knowledge, to select scores that verge towards more in or out of a set. Table 2 provides a list of all conditions. A brief contextually-based definition of each condition and calibration method is provided in SI-1 of the Supplementary Material.

*Direct Calibration:* As an example, the condition 'homogeneous sending-barangay' was determined from the percentage of households transferring collectively from the same prior barangay (local neighborhood). If nearly all households originated from the same barangay (95 percent), the relocated community was considered fully in the set. The crossover point was set at 50 percent, where no singular neighborhood group was the majority in a given site. Less than 25 percent was considered

fully out of the set, because where the largest group is 25 percent or less, the sites were composed of households from many previous barangays and no distinctly shared experience.

Similarly, geographic proximity was also calibrated with direct calibration. The condition 'close to Tacloban City center' was calibrated with anchors determined from prior relocation literature. We based proximity on distance to the city center, rather than prior communities, because households transferred from an array of prior communities with a wide range of distances. Fully in the set was defined as less than 5 km from the community to city hall, fully out of the out of the set was more than 15 km, and 10 km was set as the crossover point due to ambiguous prior findings for sites located 8–12 km away from significant economic or institutional centers [81,82].

*Indirect Calibration:* Some conditions incorporated qualitative data [83] and were calibrated indirectly. For instance, the condition 'timely permitting' relied on government documentation. Interviewees commonly suggested that developers side-stepped the permitting process, and consequently, government officials were unable to maintain oversight. Regarding NHA sites, one city government official stated, "we had minimal supervision over those settlements. In fact, they [the NHA] have railroaded everything. They did not get the necessary development permits beforehand" (2016 interview). To validate such critiques, we used the Tacloban City Planning Office status of applications for development permits as a proxy for alignment with regulatory procedures. We defined fully out of the set as a conflict between permit status and occupancy, i.e., when city-status reported no permit application had been submitted, but City Housing records indicated the site was occupied. We defined fully in the set as being issued a development permit before transferring households. Partial membership in the set was dependent on whether developers had begun the permit application process.

### 3.3.2. Analyzing and Calibrating the Outcome

Determining the membership value of each case from 0 (fully out of set membership) to 1 (fully in set membership) of livable relocation sites proceeded in several steps, as individual survey responses for numerous items were built into a composite at a community-level. Survey items were organized into thematic sub-constructs for each outcome, where built livability sub-constructs included infrastructure, housing, and accessible services, and societal livability sub-constructs included linking social capital, bonding social capital, safety, and tenure (see Table 3). In the first step, we determined individual survey respondents' membership in each sub-construct of perceived livability using partially ordered sets. In the next step, we calculated site-level scores for each sub-construct by averaging individual respondent's fuzzy-set membership scores. Averaging within each sub-construct was deemed appropriate because it lessened the influence of individuals with extreme opinions or experiences. Finally, we determined community-level scores for the higher-order constructs of built, societal, and overall livability, by assigning the minimum membership value of contained sub-constructs as the higher-order construct membership value. Taking the minimum ensures that all sub-constructs are treated as critical and was appropriate because we consider all sub-constructs to be indispensable to explaining the overarching phenomenon. In contrast, averaging or taking the maximum would mask any potentially weak sub-constructs and inflate overall livability.

**Table 2.** Specific conditions for fsQCA.

| Theorized Broad Factor | Specific Condition | General Definition for Full Set Membership | Calibration Method |
|---|---|---|---|
| Construction | Fast construction | Minimal time between Typhoon Haiyan and the date of first resident transfers. 'Fast construction' is also dwell time (the faster homes were constructed/occupied, the more time respondents have lived there prior to the survey). | *I: Case knowledge |
| | Large site | Sites were designed for 1000 households or more. | I: Case and theoretical knowledge |
| | Timely permitting | Project developers secured city-tracked development permit before construction. | I: Case knowledge |
| Community involvement | Participation | Pre-move inclusion of community members in site selection, house design, construction, and social organization. | I: Case and theoretical knowledge |
| | Cohesive transfer | Simultaneous transfer of residents into a given relocation, as opposed to staggered transfers over time. | I: Case knowledge |
| Transfer dynamics | Full occupancy | Nearly all of the constructed houses are formally occupied (by intended, not opportunistic, residents) | D |
| | Homogeneous sending-barangay | Nearly all households of a relocation community lived in the same community (barangay) before Typhoon Haiyan. | D |
| | Transitional site | Nearly all households lived in a transitional post-disaster housing project prior to relocation into permanent housing. | D |
| Site selection | Close to local hub | Minimal distance from community to intended central hub of relocation area, complete with city-government resources and local market. | D |
| | Close to Tacloban City center | Minimal distance from community to Tacloban City Hall, a proxy location for the city center. | D |

* I is indirect, D is direct.

**Table 3.** Survey items included in each sub-construct for built and societal livability.

| Built Livability Sub-Constructs | Societal Livability Sub-Constructs |
|---|---|
| *Infrastructure* | *Bonding social capital* |
| Potable water daily availability | Respondent ... trusts neighbors |
| Domestic water daily availability | Believes neighbors would share food |
| Minimal septic tank problems | Believes neighbors would help with medical needs |
| Unhindered road access into the site | Is close with neighbor |
| Daily electricity | |
| *Housing* | *Tenure* |
| Satisfaction with house | Respondent has certificate for house |
| Reported comfort | Believes they own house |
| Minimal structural defects | Believes eviction is highly unlikely |
| Adequate privacy | |
| *Accessible services* | *Safety* |
| Nearby necessities | Community is safe for young women |
| Nearby schools | Minimal concerning looking strangers |
| Nearby health centers | Communal spaces have lighting |
| | *Linking social capital* |
| | Frequency of leader-household engagement |
| | Community has successfully advocated for improvements |

In the first step, individual level fuzzy-set membership scores for perceived livability were determined using partial order theory, an algebraic and configurational toolset which leverages partially ordered sets to address ordinal data and relations without aggregating or weighting [84]. The terminology for 'partially ordered sets' is commonly simplified to 'posets' and the method the 'posetic approach'. The posetic approach was selected because it is appropriate for multi-dimensional ordinal data and allows the researcher to set qualifying thresholds for set membership, making it a natural fit with our causal analysis method, fsQCA [85–87].

When dealing with numerous ordinal items of interest, the posetic approach can help researchers manage conflicting achievements. For example, consider the survey items comprising the subconstruct infrastructure. Community A may have electricity and well-functioning septic tanks, but terrible roads, while Community B does not have electricity, but has both functional septic tanks and roadways. The state of infrastructure can thus be ordered only partially, since conflicting scores may arise, leading to incomparability. Within the posetic approach, the combined sequence of items, i.e., the responses for each of the survey items comprising infrastructure, is referred to as a 'profile'. Fuzzy-set membership for each profile is determined through a poset identification function, which quantifies the degree of membership of a profile to a researcher-defined set [88]. The researcher defines the set by establishing thresholds based off of case or theoretical knowledge. The threshold is set by selecting minimum acceptable values for each item in a profile, allowing each item to be considered in relation to the others and preventing nonsensical averaging of ordinal data. Additional details on how the posetic approach was applied to determine individual-level membership and the computational approach in R are provided in SI-2 of the Supplementary Material [89].

In the second step, we averaged individual fuzzy-set membership scores by community to determine site-level set membership for each sub-construct [88]. Averaging was deemed appropriate because numerous respondents in a given community often had the same or similar fuzzy-set membership values for a sub-construct. In the third step, we combined the sub-constructs into the higher-level constructs of built, societal, and overall livability. Methods for aggregation include taking the arithmetic average, selecting the minimum value among all items, or selecting the maximum [90]. Each option has potential drawbacks. The average suggests compensation, that one highly scoring aspect can compensate for a low-scoring aspect, yet nuanced information is lost when selecting minimum or maximum. Literature indicates dimensions of livability are essential and non-compensatory. Therefore, although we recognize tradeoffs in the selection, we used the minimum to combine categories into the higher-order construct of livability, because we consider all sub-categories to be indispensable to explaining the overarching phenomenon.

### 3.3.3. fsQCA Analysis

The analytical procedures of fsQCA can be assisted by a software of the same name, both developed by Charles Ragin [91]. The analysis relies on two key measurements, consistency and coverage. Both can be expressed as a subset relation between a condition (or combination of conditions) and the outcome. First, consistency, the degree to which cases sharing a specific condition, or combinations of conditions, agree in exhibiting the same outcome [92]. Second, coverage, the degree to which cases that share a given outcome also share a causal condition. If all (or nearly all) cases exhibiting an outcome also exhibit the condition, we can say the outcome is a subset of the condition and therefore the condition is necessary to generate the outcome. Because other means to the outcome may also exist, coverage gauges the degree to which a causal combination 'accounts for' instances of an outcome and is a measure of its empirical relevance [92]. Data analysis began with an analysis of necessary conditions for each outcome [80]. After inspecting the consistency of each individual condition, we eliminated the condition 'participation' as it displayed a consistency of less than 0.3 for each outcome.

Once the suite of conditions was selected, we built a truth table for each outcome. A truth table is a representation of the logic space of all possible combinations of conditions potentially contributing to the outcome, which can grow quite expansive for fuzzy-sets. Generally many more causal combinations are logically possible than empirically documented among cases under study [93]. In order to reduce the logic space, fsQCA allows researchers to incorporate their theorized expectations as simplifying assumptions [94]. Based off of prevailing theory, we expect the presence, not absence, of most conditions to contribute to livable relocation communities. Table S1, in SI-1 Calibration of conditions in the Supplementary Material, lists the simplifying assumptions made for each condition.

No simplifying assumptions were ever made for three conditions: fast construction, large site, and transitional site. For each of these conditions, existing literature does not definitively promote their presence or absence and we can build reasonable hypothetical narratives that either might support livability. For instance, 'fast construction', i.e., sites constructed and occupied and occupied quickly, could either indicate highly-prioritized and efficient recovery or overly rapid, low-quality development. Additionally, no simplifying assumption was made for the condition large site, because an increased housing stock and population could draw extra resources but may also be too large to maintain housing construction quality and adequately support with infrastructure services. Finally, regarding the condition 'transitional site', living in a temporary housing project prior to relocation may allow residents to bond and organize earlier, ultimately increasing their social capital and potential for advocacy. However, given the cramped and unideal conditions common in transitional sites, they also can cause rising tensions and lead to decreased social cohesion [95]. Furthermore, based on theory, we initially assumed that two conditions—close to local hub and homogeneous-sending barangay—would both would be present for livability. However, while conducting the analysis we noticed some of the most distant and heterogeneous sites achieved societal livability. We realized that such sites may have achieved high social outcomes not in spite of, but because of, a combined lack of proximity and homogeneity, and removed our assumption to allow for the analysis to better identify this pattern.

Incorporating these simplifying assumptions, we followed standard QCA practice in analyzing the intermediate solution provided by the fsQCA software's truth table minimization process [23,96]. We required a consistency cutoff of 0.8 during minimization [94]. Although raw consistency and proportional reduction in inconsistency (PRI) values rarely differed greatly, we used a rule-of-thumb of a difference greater than 0.75 to eliminate the most divergent values. Preliminary intermediary solutions were scrutinized with subset/superset analysis to identify potentially more parsimonious solutions. As the entire fsQCA process is iterative and requires several interim decisions, the accompanying Supplementary Material provides a detailed description of the decisions made for the analysis of each outcome (SI-2). Table 4 depicts the summary of values for all conditions and outcomes for each relocation community.

**Table 4.** Summary of fuzzy-set values per community.

| Community * | Close to Local Hub | Close to Tac. City Center | Cohesive Transfer | Fast Const. | Full Occ. | Homog. Sending-Brgy | Large Site | Part. | Timely Permit | Trans. Site | Built Livability | Societal Livability | Livability |
|---|---|---|---|---|---|---|---|---|---|---|---|---|---|
| Gamay | 0.93 | 0.07 | 0 | 1 | 0.97 | 0.95 | 0.67 | 0.2 | 0 | 0.26 | 0.83 | 0.70 | 0.70 |
| *Gandara* | 0.75 | 0.06 | 1 | 0.33 | 0.23 | 0.17 | 1 | 0 | 1 | 0.18 | 0.59 | 0.27 | 0.27 |
| Hitunlob | 0.93 | 0.07 | 0 | 0.67 | 0.86 | 0.30 | 0.67 | 0.2 | 0 | 0.04 | 0.77 | 0.61 | 0.61 |
| Lawigan | 0.62 | 0.19 | 0.67 | 0.67 | 0.98 | 0.94 | 0 | 0.2 | 0 | 0.96 | 0.30 | 0.82 | 0.30 |
| *Nagaja* | 0.83 | 0.02 | 0.33 | 0.33 | 0.90 | 0.13 | 1 | 0 | 0.67 | 0.12 | 0.57 | 0.61 | 0.57 |
| *Natubgan* | 0.93 | 0.06 | 1 | 0.33 | 0.92 | 0.22 | 1 | 0 | 1 | 0.10 | 0.78 | 0.29 | 0.29 |
| Oras | 0 | 0.29 | 0.33 | 0.67 | 0.31 | 0.03 | 0.33 | 0.4 | 0 | 0.08 | 0.04 | 0.72 | 0.04 |
| *Pangasugan* | 0.17 | 0.22 | 0 | 0.67 | 0.65 | 0.49 | 1 | 0 | 1 | 0.69 | 0.68 | 0.37 | 0.37 |
| Sambawan | 0.88 | 0.03 | 1 | 0.67 | 0.65 | 0.03 | 0.67 | 0.2 | 0 | 0.15 | 0.56 | 0.51 | 0.51 |
| *Suribao* | 0.87 | 0.10 | 0.33 | 0 | 0.01 | 0.10 | 1 | 0 | 0.67 | 0.06 | 0.38 | 0.20 | 0.20 |
| Ulot | 0.62 | 0.19 | 1 | 0 | 0.98 | 0.96 | 0 | 0.6 | 0.33 | 0.88 | 0.14 | 0.57 | 0.14 |
| *Villareal* | 0.94 | 0.03 | 0.67 | 0.67 | 0.92 | 0.55 | 0.33 | 0 | 1 | 0.38 | 0.81 | 0.66 | 0.66 |
| *Vulcan* | 0.38 | 0.01 | 0.67 | 0.67 | 0.69 | 0.04 | 0.67 | 0 | 1 | 0.14 | 0.11 | 0.51 | 0.11 |

* NHA communities are indicated in italics.

## 4. Results

Two conditions had low necessity for each outcome; (1) close to Tacloban City center and (2) participation. The first was a domain condition, as each site was further from the city center than we could theoretically justify as even partially in the set of 'close'. Participation had low necessity because so few organizations included participatory processes, and those that did were largely consultative or used construction labor. Notably, none of the case communities included community members in decision-making, such as where to develop the relocation site.

Out of the thirteen case communities, only a handful were partially in the set for each outcome. As predicted, not all of the cases achieving built livability also achieved societal livability, and vice versa. As such, the deconstruction of livability into built and societal dimensions allowed us to consider the pathways to different outcomes.

### 4.1. Built Livability

Eight communities were identified as in the set for built livability, indicating residents in each experienced adequate infrastructure services, housing quality, and nearby school, health centers, and basic necessities. Three pathways emerged, with six of the eight communities being covered by a pathway. Five sites managed by the government, (National Housing Authority, NHA), achieved built livability, but pathways were discerned for only three of those sites. Interestingly, sites managed by government and non-government agencies were covered by different pathways. In fsQCA notation, a condition preceded by a tilde indicates the lack of that condition, rather than its presence, is a part of the causal combination. Thus, the solutions shown in Figure 2 follow one of three pathways: (1) Large site *and* a lack of timely permitting *and* fast construction *and* close to local hub, *or* (2) Large site *and* timely permitting *and* a lack of fast construction *and* close to local hub *and* cohesive transfer, *or* (3) Large site *and* timely permitting *and* fast construction *and* not close to local hub *and* a lack of cohesive transfer.

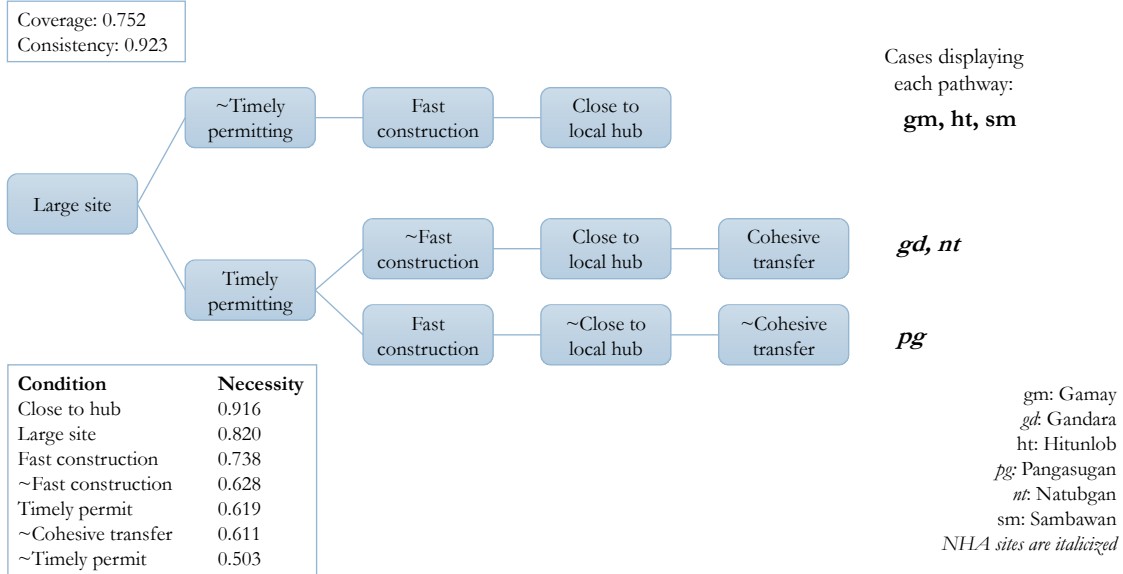

**Figure 2.** Pathways to built livability.

One condition was present in all three pathways: Large site (planned for 500 or more households). The necessity of being a large site suggests the benefits have outweighed the drawbacks of scale. Although large sites have been criticized for the concern that housing quality control may wane, larger sites likely draw more resources for improved infrastructure services. For instance, residents of the larger, NHA and NGO sites had more consistent electricity than the smallest NGO sites. Beyond large

sites, the pathways diverge into two distinct groups emerged—those that secured permits according to procedure and those that did not.

The first pathway covered three non-governmental communities, Gamay, Hitunlob, and Sambawan. One was managed by the local-office of an international organization while the other two were funded by social responsibility foundations of national corporations. None secured the city's required development permit before transferring households. Their shared pathway suggests these entities were not constrained by the institutional environment in the same way as government projects. Rather than wait for normal bureaucratic procedures such as permitting, these projects may have had informal go-aheads from the local government or may have followed more of an 'it is easier to ask forgiveness than permission' philosophy. They constructed and began moving in households quickly, necessitating the follow-on of requisite infrastructure services by other NGOs or the government. For instance, we witnessed water storage tanks donated by outside organizations and the Department of Public Works provided the city water trucks to deliver domestic water.

Permitting processes are intended to prevent ill-conceived or haphazard design, land development, and construction. That these projects succeeded in creating livable built environments without the scrutiny of permit reviewers suggests they may have used internal control processes for housing quality. At Gamay, for instance, households participated in sweat equity and had a personal hand in overseeing construction quality. The arrangement for developing both Hintunlob and Sambawan leveraged partnerships between organizations individually focused on housing construction and social development, respectively That is, a housing and social organization teamed up for Hintunlob and two different organizations formed a similar team for Sambawan. Since no one organization attempted to manage both built and social processes, engineering oversight and quality could have been better focused. Sidestepping bureaucratic process can be risky, but the sites were fortunate to be near to the local hub (less than 3 km to be considered partially in the set). This proximity made it easier to be reached by, and thus provide, infrastructure services. Because the market also houses an administrative outpost, communities near this hub may also notify the government of their grievances with their built environment more easily.

In comparison, all government (NHA) communities achieving built livability secured permits according to proper procedures, meaning NHA-contracted developers followed procedures and obtained development permits with city council before opening sites to households. Sites under the umbrella of the NHA are more institutionally constrained than the NGO sites, so it is reasonable and expected that the successful government sites secured requisite permits. The second and third pathways shared the conditions of large site and timely permitting, but contrasted in the other conditions of construction pace, proximity, and cohesiveness.

The second pathway demonstrates the cost of following procedure: proper review takes time and site development can be slow. One regional government official explained the tradeoff, "I believe in the vision of this present government, it wants things done fast, but we cannot take away from the policies that have to be adhered, the safety nets, the checks and balances" (2016 interview). Once approved, however, these sites allowed simultaneous full beneficiary move in, as indicated by the condition 'cohesive transfer'. Specifically, 'cohesive transfer' classified the synchronicity of movement into each relocation site, where a site fully in the set indicated that households moved in at approximately the same time. Cohesive move-ins may have also contributed to built livability by decreasing dormancy periods of housing units. In contrast, in communities where construction was complete, but transfers were staggered, residents used the empty houses for storage and no maintenance was done, decreasing future housing quality for newcomers.

Pangasugan differed from the other two NHA sites that achieved built livability, which resulted in a third pathway. Unlike Gandara and Natubgan, Pangasugan was constructed quickly and far from the local hub. Pangasugan was the first government site open to relocation beneficiaries and the fast pace, like with the NGO sites, may have been facilitated by buy-in from local leadership and applied extra pressure to service providers. However, construction was phased, as was beneficiary transfer. In

contrast to cohesive transfer, phased movement may allow resources and utility services to scale over time. Allowing for a lack of cohesive transfer was necessary for fast government development. While several distant NGO communities failed to be in the set for built livability, Pangasugan avoided similar struggles. Our field research revealed that government service providers prioritized government projects, and offices like the Department of Public Works and Highways worked with the NHA to provide basic services to government projects at a minimum.

### 4.2. Societal Livability

Nine of the thirteen communities were identified as in the set for societal livability, indicating residents in each experienced vibrant linking and bonding social capital, felt safe, and had a sense of tenure permanence. Four pathways emerged for societal livability, with eight out of nine communities being covered by a pathway. Six of the in-set communities are NGO projects and the remaining two are government sites. The non-covered community, Nagaja, was also an NHA site, suggesting there may be conditions contributing to social cohesion and satisfaction at government sites that we did not capture. The combined pathways for societal livability (depicted in Figure 3) are: (1) Close to local hub *and* large site *and* fast construction, or (2) Close to local hub *and* not a large site *and* cohesive transfer *and* homogeneous sending-barangay, or (3) Not close to local hub *and* fast construction *and* a lack of homogeneous sending-barangay *and* not a large site, or (4) Not close to local hub *and* fast construction *and* a lack of homogeneous sending-barangay *and* cohesive transfer.

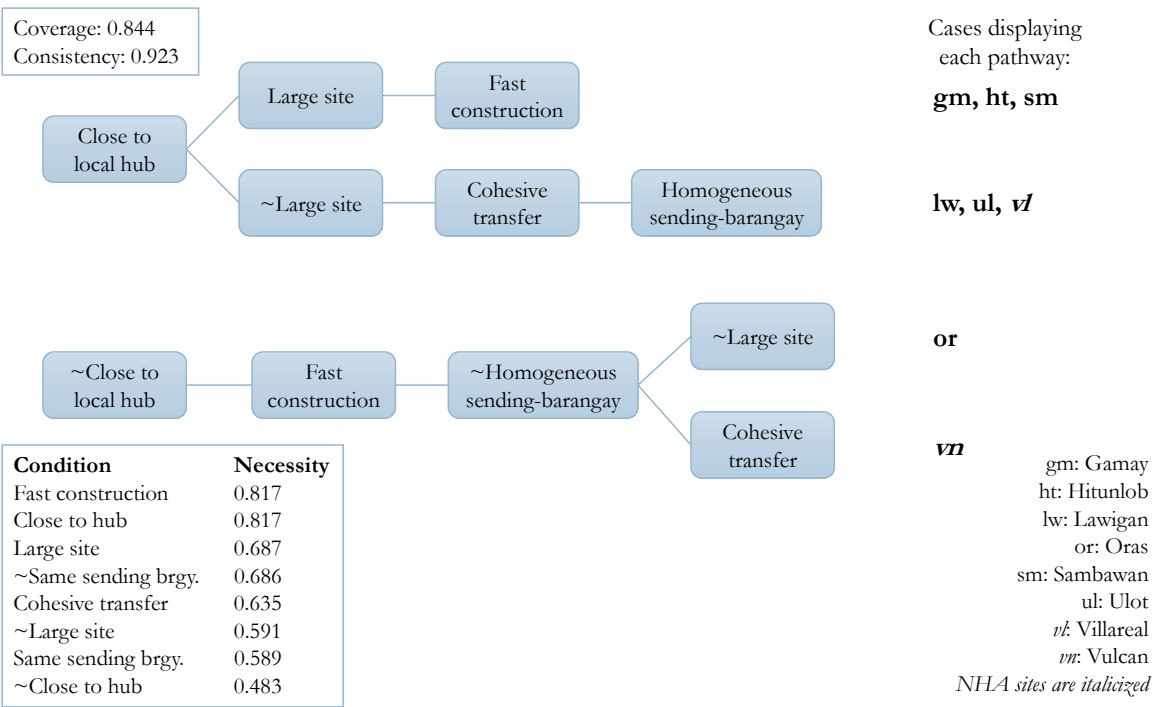

**Figure 3.** Pathways to societal livability.

Although four pathways emerged, case knowledge leads us to see three distinct types of communities achieving societal livability, summarized as (i) rapid, (ii) fully assisted or (iii) far and on their own.

The first type of community, covered by the first pathway, are those that constructed fast and near the new local hub. Projects with the fastest construction meant beneficiaries were living at the site for the longest amount of time before the survey was conducted and livability was assessed. Beneficiaries began moving into the sites covered by this pathway, Gamay, Hitunlob, and Sambawan, within a year and a half of Typhoon Haiyan. By being close and large, these communities were visible to supporting

institutions. this pathway suggests well-placed communities can develop societal livability by at around two and a half years (the timespan between their initial move-ins and the survey). Fieldwork revealed that each of these communities had community organizations and leaders active in advocacy for not only their site, but the overall relocation effort. Note, this pathway closely resembles the first pathway of built livability and covers the same three communities, further demonstrating the similarity in conditions and outcomes among them.

The second type of community, covered by the second pathway, represents the sites where beneficiaries generally all shared the same sheltering experience while transitioning from the storm to their permanent homes. At Lawigan, Ulot, and Villareal, almost all members moved into the site from transitional sites (while the condition, transitional site, was removed from the overall solution because of its low necessity for other cases, but all three sites were in the set for 'transitional site'). In particular, these cases had transitional sites designated for specific relocation sites, such that residents moved en-masse from their sending-barangay, into the transitional sites, and later cohesively from the transitional site into the relocation site. The added program management demands to support this double cohesive move likely necessitates having a smaller number of households, which we hypothesize may be why these conditions are paired with small sites on this pathway. Community members co-enduring these experiences with neighbors they had and would continue to live near likely helped foster social capital and safety. While these projects were not developed fast, the residents did not need to be in their permanent housing to begin social organization and strengthen their bonds.

These first two types of communities, rapid and fully assisted, present two considerably different ways to facilitate societal livability at relocation communities, each with different burdens for assisting organizations. The first would require enhanced engineering and construction capacity to build rapidly so that households not receiving transitional housing can move-in quickly. The latter would require devoting extra resources to community organization and support throughout a years-long, multi-stage relocation process. In comparison, the third type of community demonstrates communities thriving socially somewhat serendipitously, without the benefit of conditions we would theorize to contribute to societal livability.

The third and fourth pathways covered sites that were developed far from the local hub (recall all sites are far from the city) and with a diverse mix of households, i.e., beneficiaries from numerous different originating communities. Literature would lead us to expect this mix to preclude such sites from achieving societal livability, yet Oras and Vulcan prevailed. The pathway suggests that both sites reached societal livability not in spite of, but because of, their distance and heterogeneity. The combination of being cumbersomely far from other relocated sites and the local hub, moving in rather fast (and having time to develop relationships), and moving from different neighborhoods, worked together to support societal livability. A lack of homogeneous sending barangay was initially the most unexpected condition to emerge in the pathway, but case experience led us to understand how diversity tightened social networks at these sites because residents come to lean more on each other out of necessity. This can be illustrated with the what we witnessed as check-ins on relocated households by their old community leadership. At homogeneous sites, where a large number of occupants were transferred from the same community, we would often see vehicles branded with logos of coastal barangays, i.e., not the barangay site was currently situated in but the beneficiaries' sending barangay. For social and political reasons, the local leadership of coastal barangays maintained ties with their prior residents at sites where it was efficient to do so, those with a high proportion from their barangay. Residents of jumbled sites, especially far jumbled sites, did not experience the same check-ins. A closeness has developed at these remote communities. The mentality might be summarized as one of 'since we cannot rely on anyone else, we must rely on each other.'

*4.3. Combined Livability*

Only five of the thirteen case communities were identified as in the set for both built and societal livability. Two of the 'livable' sites were NHA communities, Nagaja and Villareal, however only

Villareal was covered by a pathway. The remainder of the livable communities were non-governmental: Gamay, Hitunlob, and Sambawan. Two pathways emerged for livability, covering all communities except Nagaja (one of the sites also not covered by a societal livability pathway). The two pathways for combined livability (depicted in Figure 4) are: (1) Close to local hub *and* fast construction *and* not a large site *and* full occupancy *and* timely permitting *and* homogeneous sending-barangay, *or* (2) Close to local hub *and* fast construction *and* large site *and* full occupancy *and* lack of timely permitting.

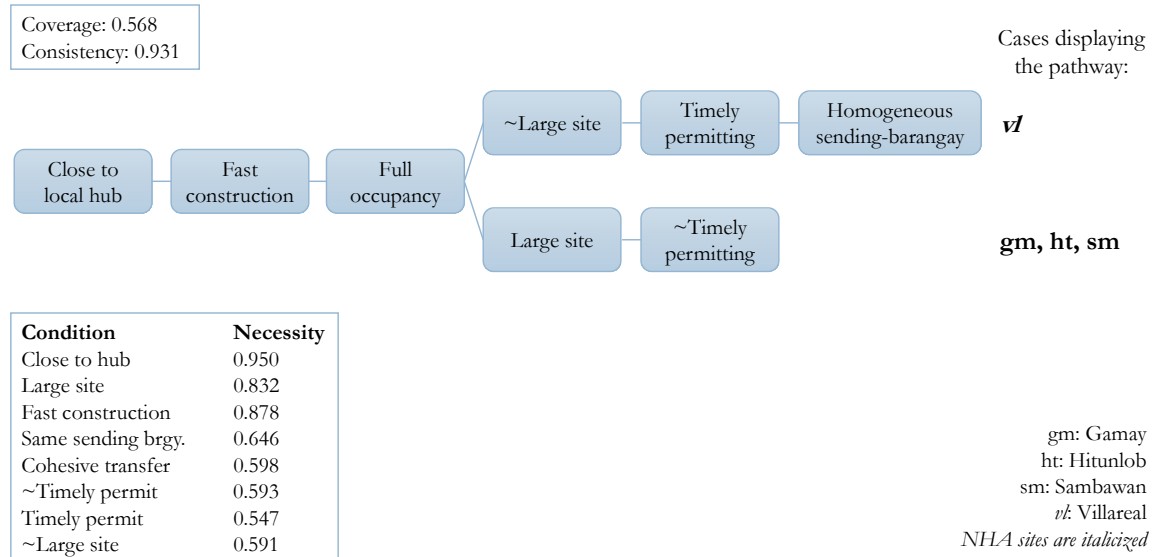

**Figure 4.** Pathway to overall livability.

On first glance, the number of conditions that combined to support overall livability is humbling, one key takeaway may be that facilitating livable relocation sites is hard. Only two of the seven NHA communities achieved both built and societal livability, indicating numerous conditions were needed for a government site to succeed. The first pathway in Figure 4 covers the NHA site of Villareal, where most beneficiaries transferred from their original community into a dedicated transitional site while they awaited the developers to secure permitting. All the while, they were nearby the local hub.

The second pathway covered the communities of Gamay, Hitunlob, and Sambawan. This pathway demonstrates that not waiting for bureaucratic permits and moving quickly can be a successful strategy for livability, when combined with proximity, size, and full occupancy. Residents in these communities experienced better livability than their relocated peers. NGOs undertaking future relocation projects should look to these three—and their combination of conditions—as the foundational typology from which to build their own programs.

In all cases, livable sites were not partially vacant. Their near full occupancy contributed to a vibrant social community and drew in resources for infrastructure services. Livable sites were also close to the local hub, which includes the Tacloban North market, a City Hall extension office, and a high school. Roadside stands have begun to crop up around the local hub, indicating the seeds of organic growth have begun to take root. In general, the necessity of being close to the local hub to achieve livability justifies a need to shift to more area-based relocation planning rather than single-site, disjointed development projects. Access to the intended institutional, economic, and educational hub matters.

Additionally, the sites achieving livability were all constructed relatively fast. Residents of these communities occupied their homes for the longest at the time of the survey suggesting that—when combined with other enabling conditions—quickly constructing new houses and transferring households helps communities get to positive outcomes faster. A note of caution may be warranted here, however. The condition of fast construction used here evaluated and compared the development pace of individual sites, not of the relocation effort as a whole. We suspect all sites would be more livable

with an extended time for diligent planning for city-wide relocation. The results here suggest only that it is important sites are constructed efficiently within the timeline of an overall relocation scheme, and do not indicate the appropriate amount of time cities and regions should take for comprehensive planning.

## 5. Limitations

There were some conditions we were not able to capture due to data access and collection constraints, such as the degree to which an agency devoted time and resources into construction oversight. Oversight may further elucidate some of the pathways, for instance the first pathway to built livability included large sites that did not follow standard permit procedures. It is unclear what supported quality construction standards, but it is possible the agencies devoted their own resources to construction oversight. Table S13 in the Supplementary Material details the excluded conditions. Furthermore, we recognize that subjective measures can be viewed not as a measure of true satisfaction but rather of toleration. Residents of the most 'livable' community in our study may tolerate objectively inferior circumstances to those in many other global communities. Nevertheless, we believe comparatively examining livability in this context allowed a deeper analysis into causation and broader investigation of post-disaster relocation.

## 6. Discussion and Conclusions

We investigated livability in post-disaster relocation communities in order to discern and compare the causal combinations of conditions leading to built and societal livability. Theoretically, we extended the literature of urban livability into post-disaster relocation housing developments. Livability has been applied in a variety of contexts, but surprisingly has yet to be operationalized as a central focus in disaster literature. As a meaningful way to identify and categorize critical outcomes, livability has the potential to conceptually bridge disaster recovery and long-term development. By including and modifying indicators targeting both the designed built and social architecture, a relocation-specific livability measure can help us understand quality of life at existing relocation projects while identifying aspects in need of improvement for future relocation projects. To do this, we used the posetic approach to assess sub-constructs of livability, allowing for ordinal, incomparable items to be systematically compared and elevated into a composite measure. The posetic approach allowed us to gather a robust qualitative understanding of life at large relocation sites, where the scale made finding theoretical saturation with interviews alone difficult. Further, we applied fuzzy-set Qualitative Comparative Analysis to identify causal pathways leading to livability. Fuzzy-set QCA facilitated the consideration of condition combinations on livability, aiding in the construction of a causal narrative.

We focused on conditions that varied not just between owner-driven and agency-driven projects, but among agency-driven projects themselves, in order to identify practical recommendations for the improvement of future (inevitable) mass relocation projects. Two conditions expected to be necessary, beneficiary participation and close proximity to the city center, were dropped from the analysis, as few projects incorporated participatory processes and none were close enough to the city center. Literature suggested adverse distance from prior neighborhoods and economic centers to be a major, even primary, drawback of relocation. Still, the findings are not neutral on the importance of proximity. The condition for distance to the new marketplace built for relocation communities emerged as central to overall livability. All five of the sites determined to be both physically and socially livable were situated nearby the local hub.

We found locating relocation sites strategically near the local economic and administrative hub aids livability, supporting the argument for improved urban planning and interconnectedness of entire relocation areas. As a result, we recommend governments and supporting organizations implementing relocation projects begin not with project-oriented planning and estimating, but area-oriented urban planning. For example, the National Housing Authority has made several updates to their standard socialized housing programs following emerging lessons learned from Typhoon Haiyan, such as the

specification that developers must ensure local governments have or will extend water service to the plots they propose for relocation development.

We recommend cities undertaking future relocation projects being by planning for the overall urban area, rather than individual project sites. This might mean selecting an appropriate site for a new 'hub' first and incentivizing (via zoning regulations, subsidized infrastructure, etc.) developers, NGOs, and other recovery actors to select nearby plots over more distant, cheaper land. Furthermore, although some sites are categorically near the local hub, none are truly interwoven. It is evident, both from an aerial examination of Tacloban North and from talking with decision makers, that there was no comprehensive street network design. Researchers have found street connectivity can impact public health and quality of life [97]. Countries in Southeast Asia have some of the highest rates of disconnected urban street networks [98]. We predict the future of relocation planning lies in thinking beyond the individual house, block, or neighborhood to address connectivity and the quality of life of the entire relocation area.

We also found livable relocation sites emerged when residents were able to bypass transitional sites and instead move quickly and fully into communities. Communities that had been fully occupied for longer enjoyed both vibrant social connections and stable infrastructure services. The bypass of transitional sites comes with important practical considerations; in-city, rental, and family-oriented housing solutions will need to be increasingly supported if governments pivot away from transitional sites for interim housing.

Overall, our methodology and findings demonstrate the potential value of characterizing variable project conditions and exploring their combined ability to lead to livability. By outlining multiple pathways, the analysis reveals complementary, not competing, strategies to promote livability. We hope this research can serve as a foundation for further systematic and comparative studies of post-disaster relocation communities.

**Supplementary Materials:** The following are available online at http://www.mdpi.com/2071-1050/12/8/3474/s1. Table S1: Summary of condition calibrations, Table S2: Close to town: Direct calibration with theoretical anchors, Figure S1: Distance from example site to City Hal (left picture) and Tacloban North Public Market (right picture), Table S3: Close to local hub: Direct calibration with case knowledge, Figure S2: Unoccupied house in 2017 (left picture) and unoccupied block in 2016 (right picture), Table S4: Full occupancy: Direct calibration with case knowledge anchors, Table S5: Barangays with largest percentage of relocated households, Table S6: Transitional site: Direct calibration with case knowledge anchors, Table S7: Transitional site: Direct calibration with theoretical anchors, Table S8: Large site: Calibration based on case knowledge, Table S9: Participation: Sum of participation components, Table S10: Cohesive transfer: Calibration based on case knowledge, Figure S3: Beneficiary transfer into each relocation site over time, Table S11: Fast construction: Calibration based case knowledge, Table S12: Timely permitting: Calibration based on case knowledge, Table S13: Postulated but excluded conditions, Table S14: Site-level summarized data for condition calibration, Table S15: Organization of sub-constructs and selected survey items, Figure S4: The posetic approach (simplified) 13 and 21 are selected as the threshold profiles (adapted from Fattore (2016)), Table S16: Threshold profiles for accessible services, Figure S5: Identification function for accessible services, Table S17: Threshold profiles for infrastructure poset, Table S18: Top ten most frequent infrastructure profiles and corresponding fs-score, Table S19: Threshold profiles for housing poset, Table S20: Most frequent housing profiles and corresponding fs-score, Table S21: Threshold profile for accessible services poset, Table S22: Most frequent accessible services profiles and corresponding fs-score, Table S23: Threshold profiles for bonding social capital poset, Table S24: Most frequent linking social capital profiles and corresponding fs-score, Table S25: Threshold profiles for bonding social capital poset, Table S26: Most frequent bonding social capital profiles and corresponding fs-score, Table S27: Threshold profiles for safety poset, Table S28: Most frequent safety profiles and corresponding fs-score, Table S29: Threshold profile for tenure poset, Table S30: Most frequent tenure profiles and corresponding fs-score, Figure S6: Explanation of set relation calculations and their associated fsQCA software terminology, Table S31: Analysis of necessity for the presence of each condition, Table S32: Analysis of necessity for the lack of each condition, Table S33: Simplifying assumptions for built livability, Table S34: Intermediate solutions for built livability, Table S35: Selected intermediate solution for built livability, Table S36: Simplifying assumptions for societal livability, Table S37: Preliminary intermediate solution for societal livability, Table S38: Intermediate solution for societal livability, Table S39: Preliminary intermediate solution for overall livability, Table S40: Intermediate solution for overall livability.

**Author Contributions:** Conceptualization, S.P. and A.J.-W.; Methodology, S.P. and A.J.-W.; Validation, S.P. and A.J.-W.; Formal Analysis, S.P.; Investigation, S.P. and A.J.-W.; Resources, S.P. and A.J.-W.; Data Curation, S.P. and A.J.-W.; Writing—Original Draft Preparation, S.P.; Writing—Review & Editing, A.J.-W.; Visualization, S.P.; Supervision, A.J.-W.; Project Administration, S.P. and A.J.-W.; Funding Acquisition, S.P. and A.J.-W. All authors have read and agreed to the published version of the manuscript.

**Funding:** This material is based upon work supported by the National Science Foundation under Grant No. 1434791 and Grant No. 1650115, and the United States Agency for International Development Office for Foreign Disaster Assistance and Habitat for Humanity International 2017 Shelter and Settlements Fellowship. Any opinions, findings, and conclusions or recommendations expressed in this material are those of the authors and do not necessarily reflect the views of the funding agencies.

**Acknowledgments:** We are indebted to the numerous survey administrators and fieldwork assistants we had the pleasure to work with in Tacloban City, including Wilma Ranes, Phoebe Tabo, Jake Delda, Lara Sudario, and Fe Callosa. Their experience and insight were invaluable. We are also grateful to the communities and community leaders who welcomed us into their neighborhoods and homes.

**Conflicts of Interest:** The authors declare no conflicts of interest.

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
