# Peer review of "Pathways to Livable Relocation Settlements Following Disaster"

_sustainability, doi:10.3390/su12083474_

Round 1

Reviewer 1 Report

Coming from a disaster, development and humanitarian background, I think that this paper make sense and present a strong case. The article is written in an accessible way and easy to understand. 

The findings are very interesting as well as useful for future post-disaster interventions. The government is obviously following a slow process due to procedures as well as necessary rules and regulations. The NGOs generally are not bound by these rules and regulations, however, they should be following the law of the land. The emerged three conditions - fast construction, full occupancy and close proximity to financial and economic centres - are very useful. This paper actually suggest a process where governments and NGOs can learn from each other. Most important, when possible each could collaborate to achieve success. 

Overall, I think that this is a solid paper with a strong contribution to the field of post-disaster reconstruction. Well done. 

Reviewer 2 Report

My  (minor) comments are listed below:

  • Use number instead of Ruth and Franklin 2014 (line number 61)
  • Thirteen relocation projects were selected. Mention the total relocation projects (line numbers 269-270)
  • Mention how many questions were included in 976 survey across the 13 relocation case sites (around line number 290)
  • Geographic proximity was determined in terms of distance to Tacloban City center. Why not in terms of orginal neighborhoods?
